# Design of a Wheel-Side Rear-Drive Distributed Electric Bus Control Strategy Based on Self-Correcting Fuzzy Control

**Wenhua Luo** [1], **Huipeng Chen** [1], **Shaopeng Zhu** [2], **Sen Chen** [1], **Jian Gao** [3], **Weiyang Wang** [1] and **Rougang Zhou** [1,4,5,*]

1    School of Mechanical Engineering, Hangzhou Dianzi University, Hangzhou 310000, China
2    Power Machinery & Vehicular Engineering Institute, College of Energy Engineering, Zhejiang University, Hangzhou 310058, China
3    Polytechnic Institute, Zhejiang University, 866 Yuhangtang Rd., Hangzhou 310058, China
4    Wenzhou Institute of Hangzhou Dianzi University, Wenzhou 325013, China
5    Mstar Technologies, Inc., Hangzhou 310012, China
*    Correspondence: zhourg@hdu.edu.cn; Tel.: +86-13345716121

**Abstract:** A suitable and effective control strategy is a prerequisite for achieving the stable driving of a distributed drive electric bus. In order to effectively utilize the advantage of the independent controllability of each rear wheel, this paper designs and compares two direct transverse moment control strategies of sliding mode control and self-correcting fuzzy control and distributes the drive torque in combination with the vehicle steering torque constraint. Moreover, based on the established seven-degrees-of-freedom vehicle model, the simulation was verified in the MATLAB/Simulink and TruckSim co-simulation platforms. The simulation results show that, compared with the sliding mode control, the self-correcting fuzzy control strategy can reduce the maximum sideslip angle deviation by 19%, 6% and 9.7%, respectively, under the double shift line condition, the high-speed small steering angle step condition and the sinusoidal line shift condition and can more effectively reduce the vehicle lateral acceleration and improve the vehicle yaw rate tracking ability, significantly improving the lateral stability of the vehicle.

**Keywords:** electric bus; yaw moment control; sliding mode control; self-correcting fuzzy control; distributed drive; lateral stability





## 1. Introduction

In recent years, with the continuous improvement of vehicle dynamic performance, vehicle driving safety has become the focus of attention. Therefore, in order to improve the driving safety of urban passenger cars, major commercial vehicle manufacturers began to pay attention to vehicle stability control technology. Due to the limitation of mechanical transmission, traditional vehicle safety control systems, such as drive skid control, anti-lock braking and vehicle stability control, have the problems of a slow response and uncontrollability. In recent years, distributed drive electric vehicles have been favored by the public due to their advantages of flexible control and a high transmission efficiency. Moreover, the new active safety system of a vehicle electronic stability system composed of direct yaw moment control (DYC) and a steering control system greatly improves the driving safety of a vehicle [1–3]. Compared with the traditional fuel vehicle, the output torque of each motor of the distributed drive electric vehicle can also be independently controlled so as to realize the electronic differential of the deflection torque of the left and right wheels, and the vehicle has a higher stability, faster response speed and higher accuracy [4,5]. At present, when studying the direct control of the vehicle deflection moment, most scholars often use the yaw rate and sideslip angle as control variables, and the commonly used methods include PID control [6,7], model predictive control [8–10], sliding mode control [11,12] and fuzzy control [13,14].

After analyzing the above literature and existing research, the PID control algorithm was determined to be simple and widely used, but it has the problem of low accuracy when solving nonlinear vehicle dynamics equations. Model predictive control can predict the future state parameters in the time domain and optimize them to improve the control accuracy. However, complex algorithms have high requirements for the performance of the controller, especially in the complex, highly nonlinear vehicle system, it is difficult to achieve real-time calculation and the application is not high. The sliding mode control algorithm is simple, has a fast response and can distribute the wheel torque in the best way under the uncertain model to maintain the stability and good robustness of the vehicle, which is suitable for the highly nonlinear vehicle system. Fuzzy control has the advantages of an independent control object and good robustness. It is especially suitable for solving problems of nonlinear models of complex vehicles, so it is also widely used in vehicle control.

Based on the literature analysis, it is found that the application prospects of sliding mode control and fuzzy control for nonlinear vehicles are good, but there is still a lot of room for improvement. In this paper, the shortcomings of the two algorithms are improved without affecting the computational complexity. For a distributed rear drive electric bus, two control strategies are designed and compared, namely, sliding mode control based on the compensating yaw moment and self-correcting fuzzy control. First, the pedal opening and steering wheel angle are input according to the driver model of TruckSim, and the expected yaw rate and centroid sideslip angle are calculated through the 2-DOF reference model of the vehicle established. Secondly, a saturation function is designed to solve the chattering problem of sliding mode control, and a self-correcting rule is designed to optimize the fuzzy control algorithm. The two improved control strategies correct the deviation by calculating the compensated yaw moment through the deviation between the expected value of the yaw rate and the actual value of the sideslip angle, respectively. The torque distribution module distributes the wheel torques according to the steering characteristics of the vehicle and the principle of maintaining neutral steering. Finally, the seven-degrees-of-freedom model of the electric bus was designed, and the co-simulation platform was built by using Matlab/Simulink and Trucksim software to compare and verify the control effects of two control strategies on the electric bus under three different working conditions: the double shift line condition, high-speed small steering angle step condition and sinusoidal line shift condition.

## 2. Build Vehicle Dynamics Models

The linear two-degrees-of-freedom model can not only describe the driver's intention better but also describe the ideal steering characteristics of the vehicle. It is simple in structure and small in computation when used to calculate the expected yaw rate and expected centroid side deflection angle. In this paper, the two-degrees-of-freedom model is used as the reference model to calculate the expected value of the yaw rate and center-of-mass sideslip angle, and a seven-degrees-of-freedom model is established to simulate and verify the influence of the control strategy on the motion of an electric bus [15,16].

### 2.1. Reference Model

Vehicle yaw stability is an important index for measuring handling stability. In order to design a high-precision control strategy and simulate the operation of the driver, a two-degrees-of-freedom vehicle model was established to obtain the expected yaw rate and sideslip angle of the vehicle. The linear two-degrees-of-freedom vehicle model, as shown in Figure 1, only considers the transverse motion along the *Y*-axis and the yaw motion around the *Z*-axis.

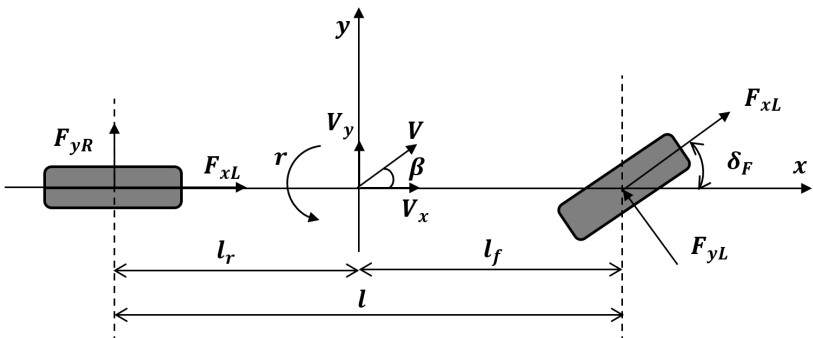

**Figure 1.** Two-degrees-of-freedom vehicle model.

In the two-degrees-of-freedom vehicle model, assuming that the influence of vehicle suspension is not considered, it is considered that the vehicle only performs simple plane motion relative to the ground and keeps the longitudinal speed of the vehicle unchanged. Ignoring the change of the tire cornering characteristics on the ground, and assuming that the turning characteristics of each tire are in the linear range, the two-degrees-of-freedom vehicle dynamics differential equation can be obtained, as shown in formula (1):

$$\begin{cases} mv_x^2\dot{\beta} = \left(k_f + k_r\right)\beta v_x + \left(l_f k_f - l_r k_r - mv_x^2\right)r - k_f v_x \delta_F \\ I_Z \dot{r} v_x = \left(l_f k_f - l_r k_r\right)\beta v_x + \left(l_f^2 k_f + l_r^2 k_r\right)r - l_f k_f v_x \delta_F \end{cases} \tag{1}$$

In the above formula, $l_f$ and $l_r$ are the distance from the center of mass of the vehicle to the front and rear axles, $l$ is the wheelbase, $k_f$ and $k_r$ are the cornering stiffness of the front and rear axles of the vehicle, $I_Z$ is the moment of inertia of the vehicle around the Z axis, $v_x$ is the longitudinal speed of the vehicle and $\delta_F$ is the front wheel steering angle of the vehicle.

There are the following equivalents when the vehicle is in a stable driving state.

$$\begin{cases} \dot{r} = 0 \\ \dot{\beta} = 0 \end{cases} \tag{2}$$

By combining formula (1) and formula (2), we can obtain:

$$\begin{cases} r_d' = \frac{v_x}{l\left(1 + Kv_x^2\right)}\delta_F \\ \beta_d' = \frac{l_r l k_r + m l_f v_x^2}{l^2 k_r \left(1 + Kv_x^2\right)}\delta_F \end{cases} \tag{3}$$

In formula (3), K is an important parameter in characterizing the steady-state response of vehicles, also known as the stability factor. The value of K can be calculated by the following formula (4):

$$K = \frac{m}{l^2}\left(\frac{l_f k_f - l_r k_r}{k_f k_r}\right) \tag{4}$$

In addition, the yaw rate and sideslip angle also need to consider that the lateral acceleration is limited by the road adhesion coefficient [17], so the boundary values of the two are shown in formula (5):

$$\begin{cases} r_{bound} = 0.85\frac{\mu g}{v_x} \\ \beta_{bound} = \tan^{-1}(0.02\mu g) \end{cases} \tag{5}$$

where $\mu$ is the road adhesion coefficient.

When the expected yaw rate and sideslip angle calculated by the linear two-degrees-of-freedom vehicle reference model exceed the maximum value provided by the road surface,

the boundary value can be selected as the expected value. By combining formula (3) and formula (5), we can obtain:

$$\begin{cases} r_d = sgn(r_d') * min(|r_d'|, r_{bound}) \\ \beta_d = sgn(\beta_d') * min(|\beta_d'|, \beta_{bound}) \end{cases} \tag{6}$$

### 2.2. Seven-Degrees-of-Freedom Vehicle Model

In this paper, a seven-degrees-of-freedom wheel rear-drive electric bus model is established to design and verify the distributed drive control strategy. In order to better obtain the vehicle running parameters and describe its stability characteristics, it is assumed that the pitch and roll motions of the vehicle are ignored, and the effects of suspension and air resistance are ignored. The vehicle is simplified as a vehicle dynamics model with seven degrees of freedom, including the longitudinal motion along the X-axis, the lateral motion along the Y-axis, the yaw motion around the Z-axis and the respective rotational motion of the four wheels. As shown in Figure 2, assuming that the wheel front and rear wheelbases are equal, a seven-degrees-of-freedom vehicle dynamics model is established.

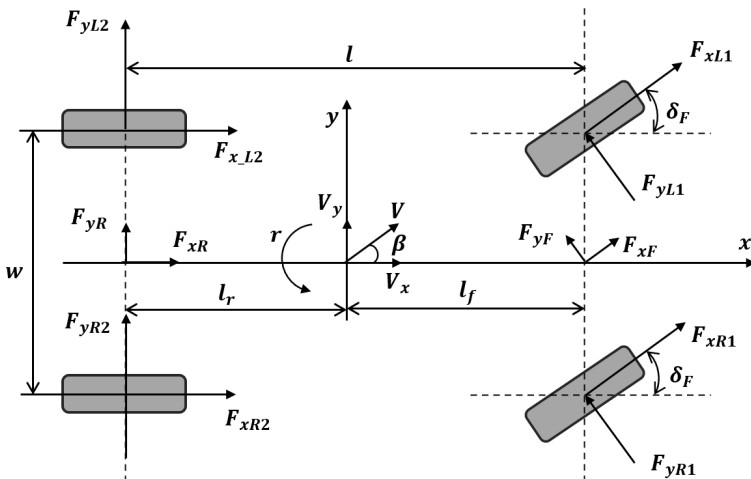

**Figure 2.** Seven-degrees-of-freedom vehicle model.

The seven-degrees-of-freedom dynamic differential equation of the vehicle can be obtained by the following formula (7):

$$\begin{cases} m\ddot{x} = m\dot{y}\dot{\varphi} + F_{x_F}cos\delta_F - F_{y_F}sin\delta_F + F_{x_R} \\ m\ddot{y} = -m\dot{x}\dot{\varphi} + F_{x_F}sin\delta_F + F_{y_F}cos\delta_F + F_{y_R} \\ I_z\ddot{\varphi} = l_f(F_{x_F}sin\delta_F + F_{y_F}cos\delta_F) + \left[(F_{x_{R1}} - F_{x_{L1}})cos\delta_F - (F_{y_{R1}} - F_{y_{L1}})sin\delta_F + (F_{x_{R2}} - F_{x_{L2}})\right]\frac{w}{2} - l_rF_{y_R} \\ I_w\dot{\omega}_R = T_d - T_b + F_dR \end{cases} \tag{7}$$

where $m$ is the mass of the vehicle; $\delta_F$ is the front wheel steering angle of the vehicle; $\varphi$ is the vehicle yaw; $r$ is the vehicle yaw rate; $I_z$ is the moment of inertia of the vehicle around the Z axis; $F_{x_F}, F_{x_R}, F_{y_F}$ and $F_{y_R}$ are, respectively, the longitudinal and lateral reaction forces of the front and rear driving wheels on the ground; $l_f$ is the distance from the center of mass to the front axle; $l_r$ is the distance from the center of mass to the rear axle; $I_w$ is the rotational inertia of the wheel; $R$ is the wheel rolling radius; $F_d$ is the wheel friction; $T_d$ is the wheel driving torque; $T_b$ is the wheel braking torque.

Nonlinear vehicle models can be built in the TruckSim control strategy simulation verification platform. In this paper, a complete electric bus model was constructed by MATLAB/Simulink and TruckSim vehicle dynamics simulation software. The co-simulation control strategy and some parameters of the electric bus in this paper are shown in Figure 3 and Table 1. The vehicle parameters in Table 1 are from the experimental vehicle data of Tianjin Tianhai Synchronization Group Co., Ltd. (Tianjin, China).

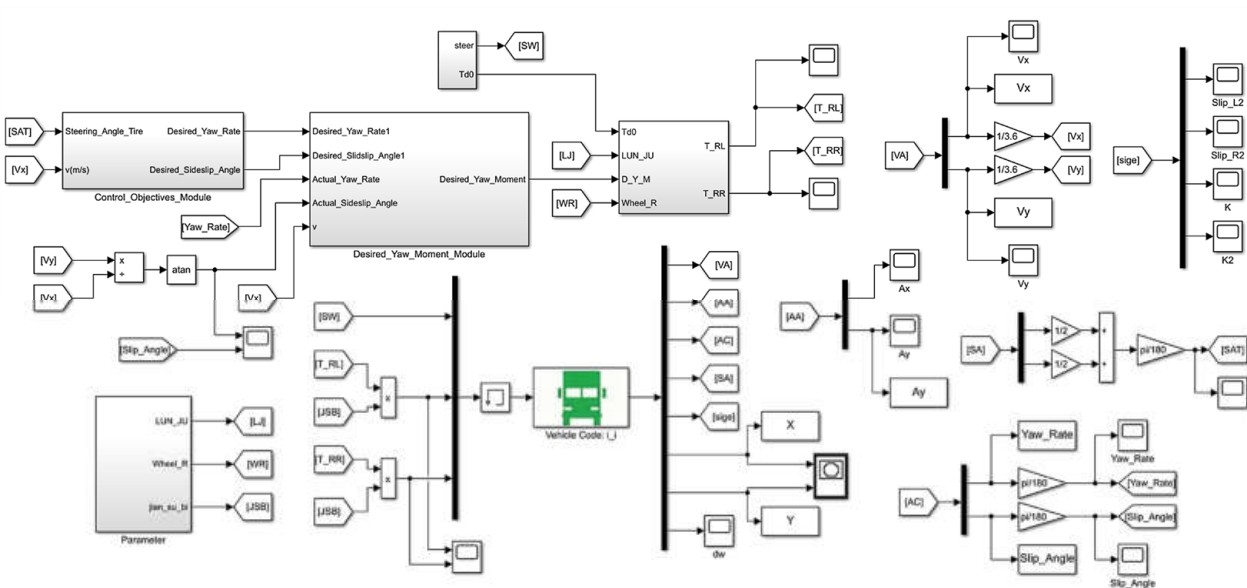

**Figure 3.** MATLAB/Simulink-TruckSim co-simulation control strategy.

**Table 1.** Reference values for some parameters of an electric bus.

| The Name of the Parameter | The Reference Value | Unit |
|---|---|---|
| Vehicle mass ($m$) | 12,800 | Kg |
| Length × Width × Height | 12,000 × 2500 × 3150 | mm |
| Height of the center mass ($h$) | 1200 | mm |
| The center of mass to the front axle distance ($l_f$) | 3240 | mm |
| The center of mass to the rear axle distance ($l_r$) | 1260 | mm |
| Wheelbase ($l$) | 4500 | mm |
| The front tire cornering stiffness ($k_f$) | 119,283.4 | N/rad |
| The rear tire cornering stiffness ($k_r$) | 225,781.4 | N/rad |
| Wheel pitch ($w$) | 1863 | mm |

*2.3. Driving Torque Distribution*

The overall driving force of the vehicle $T_d$ depends on the motor driving force and pedal opening. The driving torque distribution algorithm needs to distribute the total driving force to the two rear drive motors of the electric bus. The understeer and oversteer of the vehicle will make the inner and outer wheel torque different, resulting in front or rear axle side slip instability. In order to ensure the stable performance of the bus, this paper chooses the method of an additional yaw moment $\Delta M$ to allocate the torque of the left and right rear wheels. According to the driving characteristics of the bus, formula (8), as shown below, can be obtained:

$$\begin{cases} T_{RR} + T_{RL} = T_d \\ (T_{RR} - T_{RL})\frac{w}{2R} = \Delta M \end{cases} \tag{8}$$

where $T_{RL}$ and $T_{RR}$ are the driving torque of the left and right rear wheels, respectively, $w$ is the wheelbase of the left and right wheels and $R$ is the rolling radius of the rear wheels.

Two rear wheel drive torque distributions can be obtained after deformation:

$$\begin{cases} T_{RL} = \frac{T_d}{2} - \frac{\Delta M}{w}R \\ T_{RR} = \frac{T_d}{2} + \frac{\Delta M}{w}R \end{cases} \tag{9}$$

## 3. Distributed Drive Control System Design

In the paper, a sliding mode controller and self-correcting fuzzy controller are designed, and the influence of self-correcting fuzzy control on the vehicle control strategy is verified by comparison.

For a distributed drive electric bus, a suitable and effective control strategy can improve the stability and safety of the bus. As shown in Figure 4, considering the driving conditions of the bus, the paper puts forward a total driving torque distribution strategy based on an additional yaw moment. The strategy adopts a hierarchical structure and designs a three-layer drive control strategy. The first is the expectation setting layer, which inputs the driver information and return state parameters, formulates the expectation and inputs it to the next layer, the additional yaw moment calculation layer. According to the deviation between the actual value and the expected value of the returned state parameter, the additional yaw moment is calculated by the designed controller and output to the next layer. Finally, the total driving torque distribution layer reasonably distributes the torque output by each driving motor through an additional yaw moment and inputs it into the bus model.

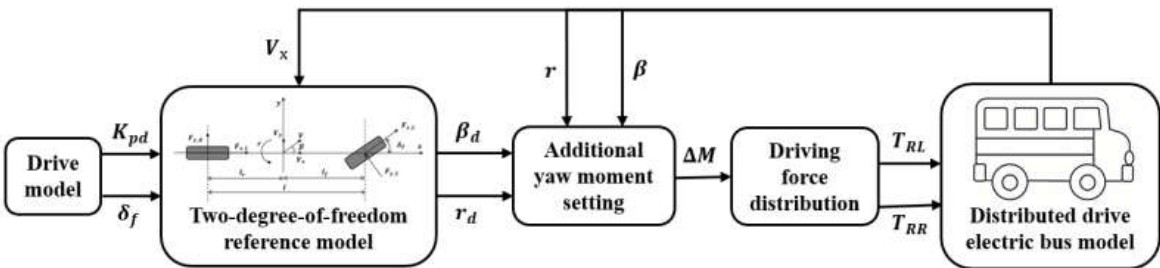

**Figure 4.** Flowchart of the Total Drive Torque Distribution Control Strategy Based on an Additional Yaw Moment.

### 3.1. Additional Yaw Moment Calculation Based on the Sliding Mode Controller

Sliding mode control is characterized by discontinuous nonlinear control, which can overcome the uncertainty of the system, has a high anti-interference ability and has a great control effect on the control of nonlinear systems.

Using the combined control of the sideslip angle and yaw rate, and adding the compensation yaw moment $\Delta M$, the two-degrees-of-freedom differential formula (1) can be transformed into formula (10):

$$\begin{cases} \dot{r} = \frac{l_f k_f - l_r k_r}{I_Z}\beta + \frac{l_f^2 k_f + l_r^2 k_r}{I_Z v_x}r - \frac{l_f k_f}{I_Z}\delta_F + \frac{\Delta M}{I_Z} \\ \dot{\beta} = \frac{k_f + k_r}{m v_x}\beta + \left(\frac{l_f k_f - l_r k_r}{m v_x^2} - 1\right)r - \frac{k_f}{I_Z}\delta_F \end{cases} \tag{10}$$

The vehicle stability control system is to ensure that the yaw rate and sideswipe angle of the vehicle have a good tracking effect on the expected value. The following selects a constant approach rate and constructs the synovial surface switching function primarily from the yaw rate (11):

$$\begin{cases} s = \lambda\left(c_r e_r + \dot{e}_r\right) + (1-\lambda)\dot{e}_\beta = \lambda c_r \dot{e}_r + \lambda\ddot{r} - \ddot{r}_d + (1-\lambda)\dot{e}_\beta \\ e_r = r - r_d \\ e_\beta = \beta - \beta_d \\ \dot{e}_r = \dot{r} - \dot{r}_d \\ \dot{e}_\beta = \dot{\beta} - \dot{\beta}_d \\ 0 < \lambda \le 1 \\ c_r > 0 \end{cases} \tag{11}$$

where $s$ is the sliding mode variable of the controller; $\lambda$ is the weight coefficient; $c_r$ is the relative weight coefficient between the yaw rate deviation and the derivative.

When the control input of the regulating system keeps it moving on the sliding surface and the system tends to be stable, $s = \dot{s} = 0$. Then, by combining formula (10) and formula (11), we can obtain:

$$\Delta\dot{M} = -I_Z\left(c_r\dot{e}_r + \frac{1-\lambda}{\lambda}\dot{e}_\beta + \frac{l_fk_f - l_rk_r}{I_Z}\dot{\beta} + \frac{l_f{}^2k_f + l_r{}^2k_r}{I_Zv_x}\dot{r} - \frac{l_fk_f}{I_Z}\dot{\delta}_F - \ddot{r}_d + K_vsgn(s)\right) \tag{12}$$

where $sgn(s)$ is a symbolic function, indicating that the return value is $-1$ when $s < 0$, 0 when $s = 0$ and 1 when $s > 0$. $K_v$ is a constant approaching rate, so it is necessary to ensure that $K_v > 0$. The value of $K_v$ determines the response time of the system to make the control system have practical value, and the value of $K_v$ should not be too small. In the actual sliding mode control, the ideal switching characteristics cannot be established. Considering the inertia of the system and the discreteness of the communication message, it is necessary to lag control over time and space. The large $K_v$ makes the state trajectory of the system unable to completely slide to the equilibrium point along the designed sliding mode surface, but it constantly crosses back and forth on both sides of the sliding surface, which leads to the chattering phenomenon in general sliding mode control. This is an inevitable shortcoming in ordinary sliding mode control, and the chattering problem will not only reduce the control accuracy but also increase the energy consumption and accelerate the wear of vehicle components. Therefore, the chattering problem needs to be controlled.

In order to eliminate or alleviate this jitter, the commonly used schemes are the filtering method [18], genetic algorithm optimization method [19], reducing switching gain method [20,21] and so on. In this paper, we choose the scheme of adding the boundary layer method, that is, the $sgn(s)$ symbolic function in formula (13) is replaced by the saturation function, and the $sat(s)$ saturation function is shown in formula (13):

$$sat(s) = \begin{cases} 1, & s > \Delta_r \\ \gamma s, & |s| \leq \Delta_r, \gamma = \frac{1}{\Delta_r} \\ -1, & s < -\Delta_r \end{cases} \tag{13}$$

where $\Delta_r$ is the boundary layer thickness parameter, which satisfies the constraint condition of the boundary layer thickness $\Delta_r > 0$. When $s$ is on the inside of the boundary layer, the function changes linearly, while when $s$ is outside the boundary layer, the value of the original $sgn(s)$ symbolic function is maintained by the saturation setting. Then, formula (12) becomes the form of the following formula (14):

$$\Delta\dot{M} = -I_Z\left(c_r\dot{e}_r + \frac{1-\lambda}{\lambda}\dot{e}_\beta + \frac{l_fk_f - l_rk_r}{I_Z}\dot{\beta} + \frac{l_f{}^2k_f + l_r{}^2k_r}{I_Zv_x}\dot{r} - \frac{l_fk_f}{I_Z}\dot{\delta}_F - \ddot{r}_d + K_vsat(s)\right) \tag{14}$$

By integrating formula (14), the compensated yaw moment $\Delta M$ can be obtained.

Finally, we also need to test the stability of the joint control system of the yaw rate and centroid yaw angle. We define the Lyapunov function, as shown in formula (15).

$$L = \frac{1}{2}s^2 \tag{15}$$

According to the principle of chain derivation, there is:

$$\dot{L} = s\dot{s} = s\left(\lambda\left(c_r\dot{e}_r + \frac{l_fk_f - l_rk_r}{I_Z}\dot{\beta} + \frac{l_f{}^2k_f + l_r{}^2k_r}{I_Zv_x}\dot{r} - \frac{l_fk_f}{I_Z}\dot{\delta}_f + \frac{\Delta\dot{M}}{I_Z} - \ddot{r}_d\right) + (1-\lambda)\dot{e}_\beta\right) \tag{16}$$

Substitute formula (14) into formula (16); then,

$$\dot{L} = s\dot{s} = s(-K_vsat(s)) = \begin{cases} -K_v|s|, & |s| > \Delta_r \\ -K_v\gamma s^2, & |s| \leq \Delta_r \end{cases} \tag{17}$$

According to the previous definition, the average values of $K_v$ and $\gamma$ are greater than 0, so $\dot{L} \leq 0$ is always true. As a result, the stability effect of the designed control system is shown.

### 3.2. Additional Yaw Moment Calculation Based on the Self-Correcting Fuzzy Controller

The ordinary fuzzy controller mainly includes three modules: fuzzy input, fuzzy reasoning and anti-fuzzy output [22]. In the whole fuzzy control system, the modification of control rules and the membership function has a great influence, while the selection of the scale factor has greater flexibility. Moreover, it is very important to transform the basic domain and fuzzy set domain by the scale factor to stabilize the control system and other performance indexes.

Therefore, the self-correcting fuzzy controller designed in this paper is based on the ordinary fuzzy control, adding the function of a self-correcting scale factor. Without calculating the performance index, the three scaling factors are adjusted simultaneously online by directly using the feature information provided by the dynamic process of the system. As shown in Figure 5, one of the input variables of the self-correcting fuzzy controller is the deviation $e_r$ between the actual value and the expected value of the yaw rate, and the other input variable is the deviation $e_\beta$ between the actual value and the expected value of the sideslip angle. The output variable is the additional yaw moment $\Delta M$, $K1$ and $K2$ are the proportional factors of the input variables and $K3$ is the proportional factor of the output variable.

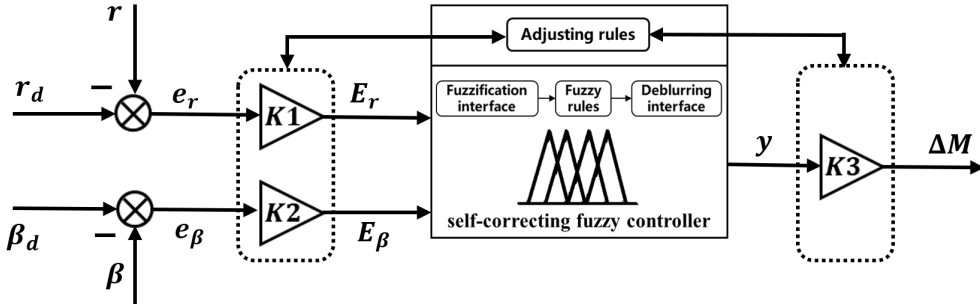

**Figure 5.** Additional yaw-moment calculation based on the self-correcting fuzzy controller.

In Figure 5, the following relationship can be known:

$$\begin{cases} E_r = e_r \cdot K1 \\ E_\beta = e_\beta \cdot K2 \\ \Delta M = y \cdot K3 \end{cases} \tag{18}$$

The fuzzification process is to transform the variables into the corresponding fuzzy set domain [23]. Let the fuzzy set domain after the two input variables and one output variable be changed, as shown in the formula (19):

$$\begin{cases} T(E_r) = \left\{ A_r^1, A_r^2, \cdots, A_r^m \right\} \\ T(E_\beta) = \left\{ A_\beta^1, A_\beta^2, \cdots, A_\beta^m \right\} \\ T(y) = \left\{ B^1, B^2, \cdots, B^m \right\} \end{cases} \tag{19}$$

The corresponding membership functions are:

$$\begin{cases} \left\{ \mu_{A_r^1}(E_r), \mu_{A_r^2}(E_r), \cdots, \mu_{A_r^m}(E_r) \right\} \\ \left\{ \mu_{A_\beta^1}(E_\beta), \mu_{A_\beta^2}(E_\beta), \cdots, \mu_{A_\beta^m}(E_\beta) \right\} \\ \left\{ \mu_{B^1}(y), \mu_{B^2}(y), \cdots, \mu_{B^m}(y) \right\} \end{cases} \tag{20}$$

The input adopts the fuzzy method of a single-point fuzzy set. Through fuzzy reasoning, the rule adaptation function and the membership function of the fuzzy set output by each rule can be obtained, as shown in formula (21):

$$
\begin{cases}
\alpha_i = \mu_{A_r^i}(E_r)\mu_{A_\beta^i}\left(E_\beta\right) \\
\mu_{B^j}(y) = \alpha_i \mu_{B^j}(y)
\end{cases}
\tag{21}
$$

Taking the weighted average method as the anti-fuzzification method, the output can be obtained as:

$$
\begin{cases}
y = \dfrac{\sum_{i=0}^m y_{c_i}\mu_{B_i}\left(y_{c_i}\right)}{\sum_{i=0}^m \mu_{B_i}\left(y_{c_i}\right)} \\
\mu_{B_i}(y_{c_i}) = max\left\{\mu_{B_i}(y)\right\} = \alpha_i
\end{cases}
\tag{22}
$$

where $y_{c_i}$ is the maximum point taken by $\mu_{B_i}(y)$, which is generally the center point of the membership function.

Therefore, the output expression is:

$$
y = \frac{\sum_{i=0}^m y_{c_i}\alpha_i}{\sum_{i=0}^m \alpha_i} = \frac{\sum_{i=0}^m y_{c_i}\mu_{A_r^i}(K1*e_r)\mu_{A_\beta^i}\left(K2*e_\beta\right)}{\sum_{i=0}^m \mu_{A_r^i}(K1*e_r)\mu_{A_\beta^i}\left(K2*e_\beta\right)}
\tag{23}
$$

By substituting formula (23) into formula (18), the additional yaw moment $\Delta M$ can be obtained, as shown in formula (24):

$$
\Delta M = \frac{\sum_{i=0}^m y_{c_i}\mu_{A_r^i}[K1*e_r]\mu_{A_\beta^i}\left[K2*e_\beta\right]}{\sum_{i=0}^m \mu_{A_r^i}[K1*e_r]\mu_{A_\beta^i}\left[K2*e_\beta\right]} * K3
\tag{24}
$$

The corresponding fuzzy rules of the described input–output relation take the following form:

$$
R_i: \ if \ e_r = A_r^i \ and \ e_\beta = A_\beta^i \ then \ \Delta M = B^i, \ and \ i = (1,2,\cdots,m)
$$

where $m$ is the total number of fuzzy rules.

From formula (24), it can be seen that the $\Delta M$ not only depends on the input deviations $e(r)$ and $e(\beta)$ but is also affected by the quantization factor $K1$, $K2$ and the scale factor $K3$. In the general fuzzy control, the value of the quantization factor $K1$, $K2$ and the scale factor $K3$ will not be changed after the determination, so the output of the controller $\Delta M$ has a small range of change, resulting in a small adjustment range of the yaw moment when the bus is running, which is unable to adapt to all the bus driving conditions.

Obviously, choosing different scale factors has a great influence on the control effect of the fuzzy control system.

When the control rules and membership functions are determined, the quality of the control loop can be greatly improved by adjusting the scale factor. Now, the rule of scale factor is adjusted as follows:

$$
R_i: \ if \ e_r = A_r^i \ and \ e_\beta = A_\beta^i \ then \ d(K1_i) = B^i, \ and \ i = (1,2,\cdots,m)
$$

where, $m$ is the total number of fuzzy rules.

The modified $d(K1_i), d(K2_i)$ and $d(K3_i)$ can be obtained or corrected by using the same or opposite fuzzy reasoning and fuzzy resolution methods, so formulas (25)~(27) are obtained [24]:

$$
K1_{i+1} = K1_i + \delta 1 * d(K1_i)
\tag{25}
$$

$$
K2_{i+1} = K2_i + \delta 2 * d(K2_i)
\tag{26}
$$

$$
K3_{i+1} = K3_i + \delta 3 * d(K3_i)
\tag{27}
$$

where $\delta 1$, $\delta 2$ and $\delta 3$ are the correction coefficients, which are used to control the correction coefficients. In this paper, the fuzzy controller fuzzified the precise values of the input deviations $e_r$ and $e_\beta$, as well as the quantization factor $K1$, $K2$ and the scale factor $K3$, into five fuzzy sets, which are Negative Big (NB), Negative Small (NS), Zero (ZE), Positive Small (PS) and Positive Big (PB). The output variable $\Delta M$ is divided into seven fuzzy sets, which are Negative Big (NB), Negative Medium (NM), Negative Small (NS), Zero (ZE), Positive Small (PS), Positive Medium (PM) and Positive Big (PB). The division of fuzzy sets designed in this paper is shown in Table 2. Figures 6–8 show the corresponding membership functions.

**Table 2.** Fuzzy set partition table.

| $e_r$ | $e_\beta$ | $d(K1_i)$/ $d(K2_i)$/$d(K3_i)$ | $\Delta M$ |
|---|---|---|---|
| NB | NB | NB | NB |
| NS | NS | NS | NM |
| ZE | ZE | ZE | NS |
| PS | PS | PS | ZE |
| PB | PB | PB | PS |
| | | | PM |
| | | | PB |

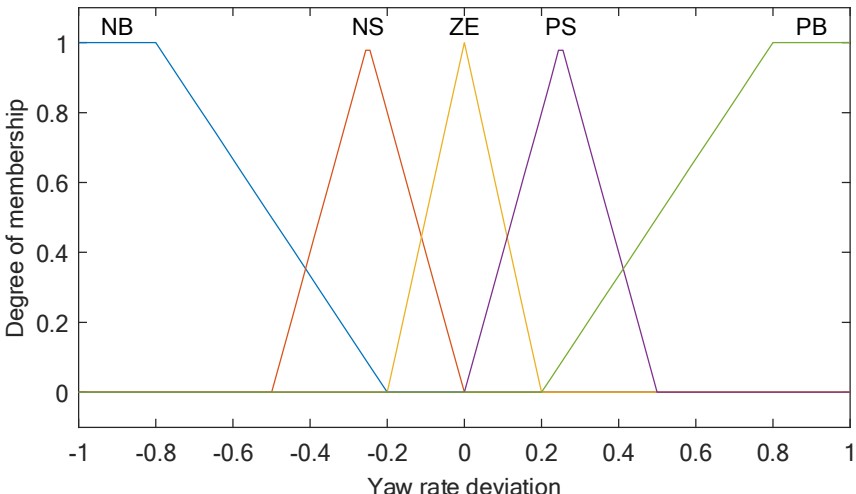

**Figure 6.** Membership function of the yaw rate deviation.

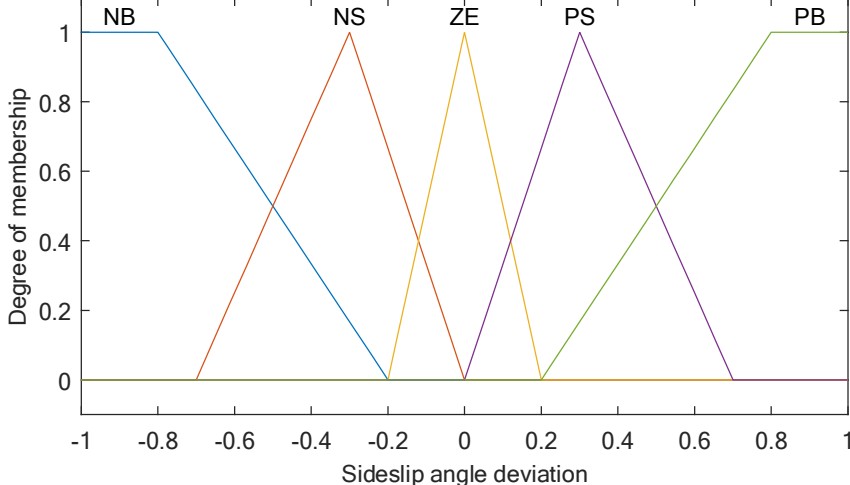

**Figure 7.** Membership function of the side slip angle deviation.

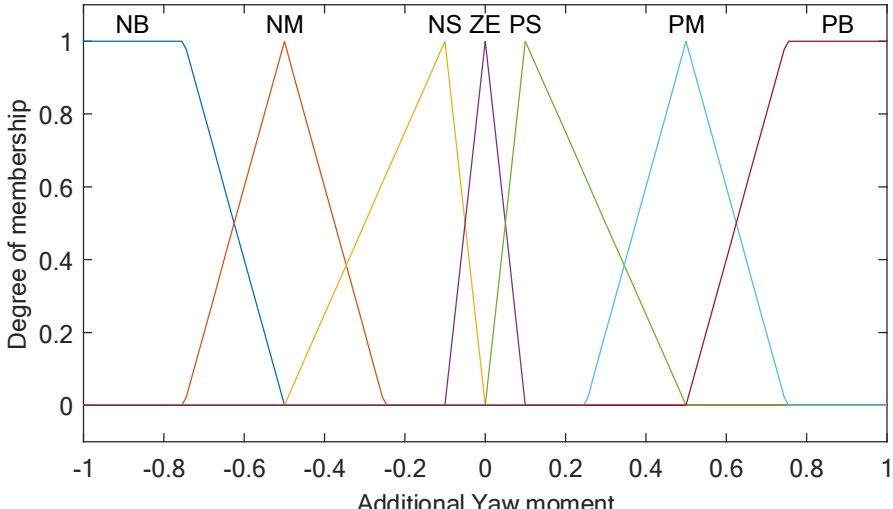

**Figure 8.** Membership function of the additional yaw moment.

Fuzzy reasoning is the core link in fuzzy control, which is to form a one-to-one rule relationship between the input variable combination of the fuzzy controller and the output variable based on expert experience.

The self-correction of K1, K2 and K3 is realized according to different driving conditions of the vehicle. When the vehicle is running at low speeds and turning, the yaw rate has great control, so it is necessary to appropriately increase the quantization factor K1 and increase the scaling factor K3 to improve the response speed. When the vehicle is running at a medium-high speed and turning, the yaw rate at the small corner is equivalent to the control weight of the sideslip angle, and the quantization factors K1 and K2 need to be increased. K3 should be reduced to maintain the driving stability. At the large corner, the sideslip angle needs to be controlled first. The quantization factor K2 needs to be increased appropriately, and the scaling factor K3 needs to be reduced to ensure the stable running of the vehicle [25].

The adjustment rules and global fuzzy logic rules used in this paper are shown in Tables 3 and 4, each with 25 fuzzy rules.

**Table 3.** Adjustment rules Fuzzy Rule Table.

| $e_r \backslash e_\beta$ | **NB** | **NS** | **ZE** | **PS** | **PB** |
|---|---|---|---|---|---|
| **NB** | NB | NS | PS | NS | NB |
| **NS** | NB | PS | ZE | PS | NB |
| **ZE** | NB | ZE | ZE | ZE | NB |
| **PS** | NB | PS | ZE | PS | NB |
| **PB** | NB | NS | PS | NS | NB |

**Table 4.** Fuzzy Control Fuzzy Rule Table.

| $e_r \backslash e_\beta$ | **NB** | **NS** | **ZE** | **PS** | **PB** |
|---|---|---|---|---|---|
| **NB** | NB | NB | NB | NM | NM |
| **NS** | NB | NM | NM | NS | NS |
| **ZE** | NS | NS | ZE | PS | PS |
| **PS** | PS | PS | PM | PM | PB |
| **PB** | PM | PM | PB | PB | PB |

## 4. Simulation Analysis

Based on the distributed rear-wheel drive electric bus model, three typical driving conditions—a double shift line condition, a high-speed small-steering-angle step condition

and a sinusoidal line shift condition—are selected to verify and analyze the control effect of the designed distributed drive control strategy. At the same time, the bus running states of the three control schemes under the control of no control, sliding mode control and self-correcting fuzzy control are compared. According to the above analysis and design, the Matlab/Simulink software (Matlab R2016b, MathWorks, Natick, Massachusetts, USA) and TruckSim software (TruckSim 2020, Mechanical Simulation Corporation, Ann Arbor, Michigan, United States) are used to establish a co-simulation platform, and the block diagram of the control strategy is shown in Figure 9.

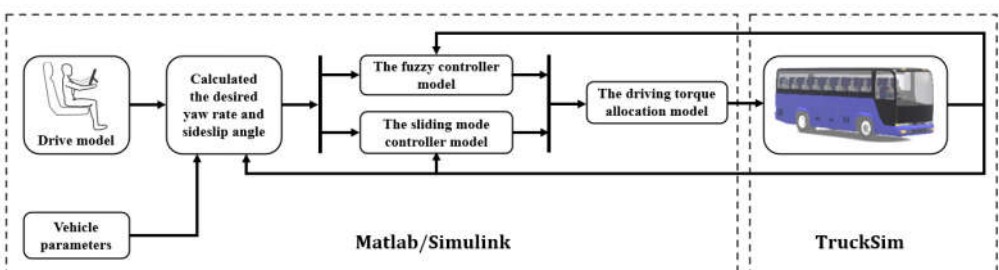

**Figure 9.** Matlab/Simulink–TruckSim co-simulation platform.

### 4.1. Double Shift Line Condition

The double line shifting condition is generally used to simulate the process of overtaking or the emergency avoidance process when encountering obstacles, which is a typical working condition of the closed-loop system. The road diagram of the double lane change condition is shown in Figure 10. In this paper, the double shift line condition is selected to simulate an emergency turn at a large corner at medium speed. It is assumed that the road adhesion coefficient $\mu = 0.7$ and the initial driving speed $v = 50 \ km/h$, and the steering wheel angle signal is shown in Figure 11. The driver rotates the steering wheel to change lanes after 5 s and then rotates the steering wheel back to the main road. The longitudinal speed curve, yaw rate curve, lateral acceleration curve, wheel torque output curve and sideslip angle curve are shown in Figure 12. The maximum values of the main control parameters under the double shift line condition are shown in Table 5.

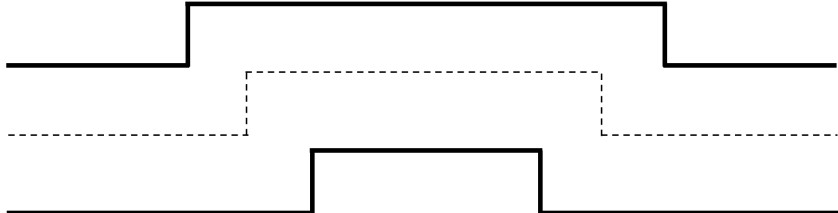

**Figure 10.** Road diagram of the double shift line condition.

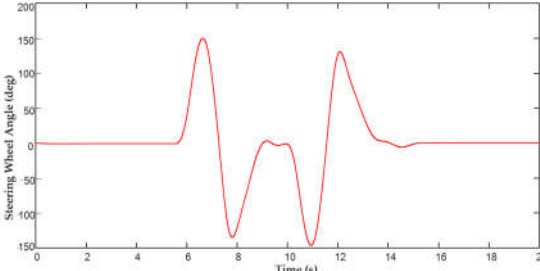

**Figure 11.** Steering wheel angle at the double shift line condition.

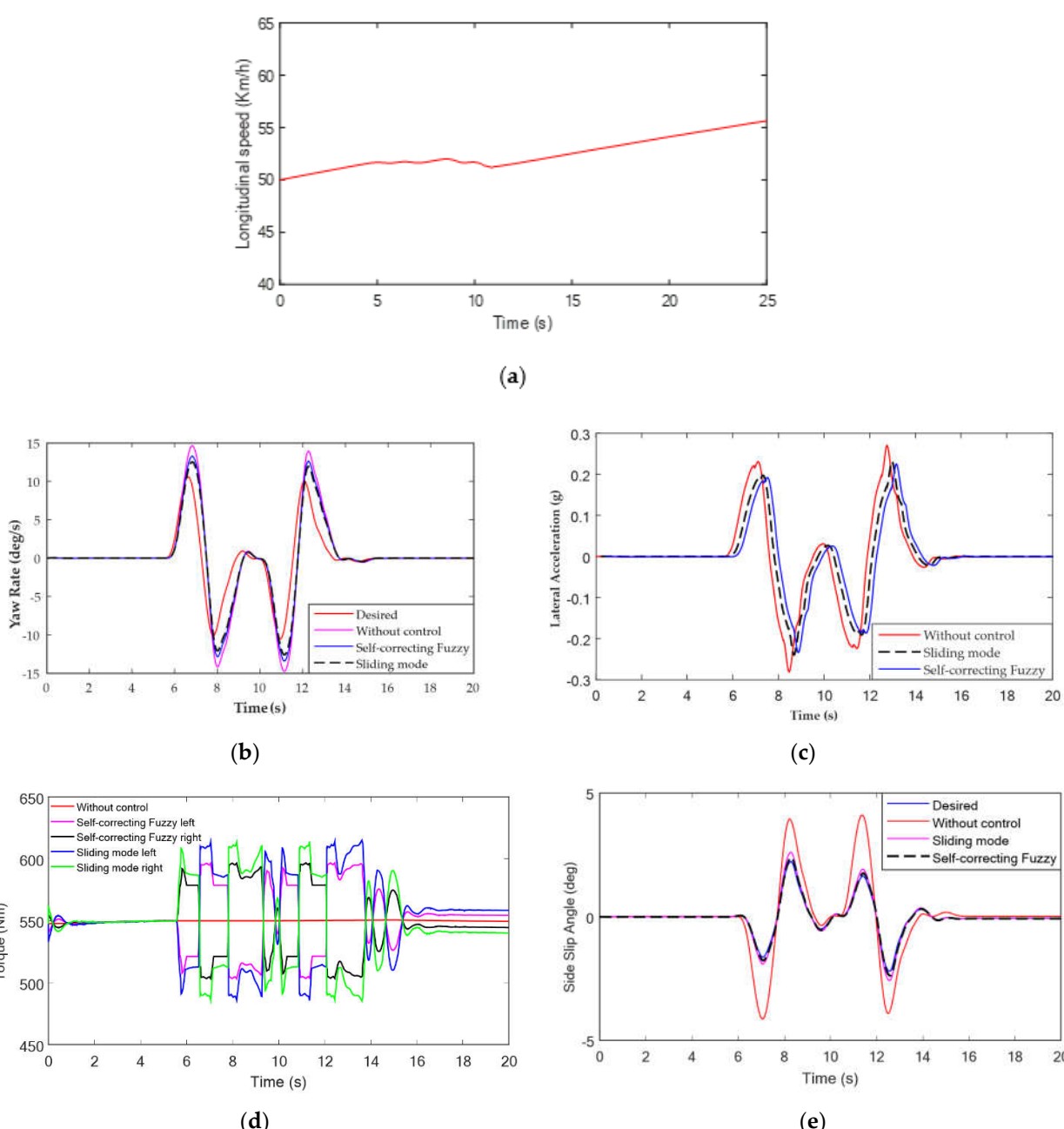

**Figure 12.** Simulation results of the distributed drive control at the double shift line condition: (**a**) Speed; (**b**) Yaw Rate; (**c**) Lateral Acceleration; (**d**) Torque; (**e**) Sideslip Angle.

**Table 5.** The maximum values of the main control parameters under the double shift line condition.

| Control Strategy | Without Control | Sliding Mode | Deviation (%) | Self-Correcting Fuzzy | Deviation (%) |
|---|---|---|---|---|---|
| Maximum yaw rate (deg/s) | 14.72 | 12.21 | 22 | 13.02 | 30 |
| Maximum sideslip angle (deg) | 4.11 | 2.59 | 28 | 2.20 | 9 |
| Maximum lateral acceleration (g) | 0.283 | 0.224 | / | 0.219 | / |

As shown in Figure 12 and Table 5, in the simulation of the double shift line condition, the longitudinal speed of the vehicle is basically unchanged. For uncontrolled vehicles, the actual values of the yaw rate and sideslip angle are much higher than the expected values. The maximum yaw rate reaches 14.72 deg/s, and the maximum sideslip angle

reaches 4.11 degrees. The vehicle trajectory deviates seriously from the expected trajectory and is prone to instability. Under sliding mode control and self-correcting fuzzy control, the vehicle's yaw rate and centroid yaw angle can follow the expected value well, the output torque difference between the left and right sides of the wheel is large and the electronic differential control effect is obvious. Under the sliding mode control strategy, the maximum yaw angular velocity and the centroid yaw angle are 12.21 deg/s and 2.59 degrees, respectively. Under the self-correcting fuzzy control strategy, the maximum values of the yaw rate and the centroid yaw angle are 13.02 deg/s and 2.20 degrees, respectively, and the torque fluctuation between the two wheels is smaller, so the vehicle is more comfortable at medium to high speeds.

The simulation results show that the conventional sliding mode control strategy and self-correcting fuzzy control strategy can ensure that the vehicle can track the expected trajectory and that the tracking effect is good when the vehicle runs at a medium speed and large angle under the double shift line condition compared with the condition without control. However, the deviation between the actual value and the expected value of the maximum centroid side angle of the vehicle under the self-correcting fuzzy control is about 19% smaller than that under the sliding mode control, which indicates that the self-correcting fuzzy control strategy can better control the centroid side angle and improve the steering stability under this working condition. At the same time, the lateral acceleration of the vehicle under the self-correcting fuzzy control is smaller and gentler than that under the sliding mode control, which indicates that the vehicle runs more stably in the case of emergency avoidance and lane overtaking, which improves the driving safety of the vehicle. At the same time, it is proved that the designed self-correcting fuzzy control can achieve a good control performance of the centroid side deflection angle under the conditions of a medium speed and a large angle, and the effectiveness of the self-adjusting rule under this condition is verified.

### 4.2. High-Speed Small-Steering-Angle Step Condition

This condition simulates the vehicle accelerating at a higher speed on the road with a smaller steering angle, which is generally used to test the transient response performance of the vehicle. Assume that the road adhesion coefficient $\mu = 0.7$ and the initial speed $v = 80$ km/h. The steering wheel angle signal is shown in Figure 13, and the accelerator pedal signal is shown in Figure 14. The driver starts to rotate the steering wheel by 50 degrees from 6 s and starts to step down on the accelerator pedal after 10 s to accelerate. The longitudinal speed curve, the torque output curve of each wheel, the yaw rate curve, the sideslip angle curve and the lateral acceleration curve are shown in Figure 15. The maximum values of the main control parameters under the high-speed small-steering-angle step condition are shown in Table 6.

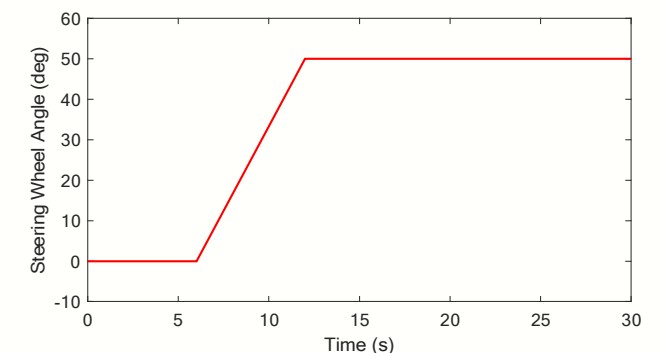

**Figure 13.** Steering wheel angle at a high-speed small-steering-angle step condition.

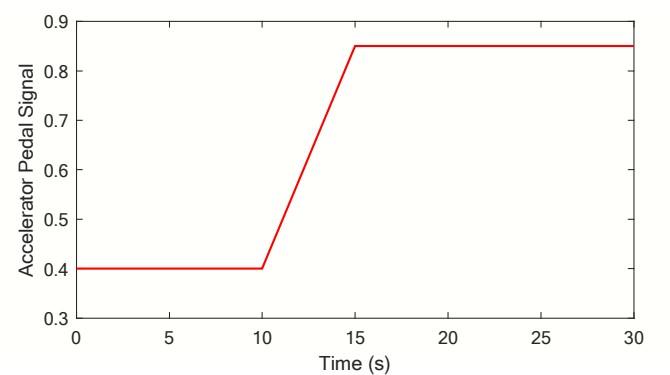

**Figure 14.** Acceleration pedal at a high-speed small-steering-angle step condition.

**Figure 15.** Simulation results of the distributed drive control at a high-speed small-steering-angle step condition: (**a**) Speed; (**b**) Torque (**c**) Yaw Rate; (**d**) Sideslip Angle; (**e**) Lateral Acceleration.

**Table 6.** The maximum values of the main control parameters under the high-speed small-steering-angle step condition.

| Control Strategy | Without Control | Sliding Mode | Deviation (%) | Self-Correcting Fuzzy | Deviation (%) |
|---|---|---|---|---|---|
| Maximum yaw rate (deg/s) | 5.53 | 4.96 | 30 | 4.53 | 19 |
| Maximum sideslip angle (deg) | 2.42 | 2.05 | 21 | 1.95 | 15 |
| Maximum lateral acceleration (g) | 0.34 | 0.3 | / | 0.275 | / |

As shown in Figures 13–15, from the 6th second to the 12th second, the vehicle enters the steering process, and the steering angle gradually increases to 50 degrees. At this time, the vehicle speed is basically unchanged, but the torque of the left and right wheels starts to change, and the yaw rate of the vehicle and the error value of the sideslip angle of the center of mass start to change. After 12 s, the steering angle remained at 50 degrees, and the accelerator pedal opening gradually increased to 0.85 at the 15th second. The speed increased during acceleration, and the deviation between the actual value and the expected value of the yaw speed began to increase. As shown in Figure 15 and Table 6, for uncontrolled vehicles, the actual values of the yaw rate and sideslip angle are much greater than the expected values, and the vehicle has a serious tendency to oversteer, is prone to sideslip and drift and is in a very dangerous driving state. Under the sliding mode control strategy, the maximum yaw rate, the maximum sideslip angle and the maximum lateral acceleration of the vehicle are reduced to 4.96 deg/s, 2.05 degrees and 0.3, but they are all greater than the maximum control parameters of the self-correcting fuzzy strategy. Under the self-correcting fuzzy control strategy, the actual value of the maximum yaw rate is 4.53 deg/s, the actual value of the maximum sideslip angle is 1.95 degrees, which is closest to the expected value, and the maximum lateral acceleration is 0.275, which can better restrain the roll.

The simulation results show that, compared with no control, both the self-correcting fuzzy control and the sliding mode control can track the expected value well under the high-speed and small-angle step condition. Under this condition, the yaw rate and centroid sideslip have a great influence on vehicle stability. Compared with sliding mode control, the self-correcting fuzzy control strategy can reduce the maximum yaw rate deviation of 11% and the maximum centroid sideslip deviation of 6%, indicating that the self-correcting fuzzy control can better ensure the tracking effect of the vehicle and restrain the trend of oversteering. At the same time, compared with the sliding mode control strategy, the maximum lateral acceleration under the self-correcting fuzzy control strategy is also reduced by 0.25, which improves the stability and comfort of the vehicle and also proves the effectiveness of the fuzzy adjustment rule in this working condition.

*4.3. Sinusoidal Line Shift Condition*

This working condition simulates a situation in which the vehicle performs a large lane change at a medium speed on a low-adhesion road. Assume that the road adhesion coefficient $\mu = 0.3$ and the initial speed $v = 50$ km/h. The steering signal is shown in Figure 16, and the accelerator pedal signal is shown in Figure 17. The steering wheel performs a 120-degree reciprocating steering operation after 4 s. The vehicle longitudinal speed curve, wheel torque output curve, yaw rate curve, sideslip angle curve and lateral acceleration curve are shown in Figure 18. The maximum values of the main control parameters under the sinusoidal line shift condition are shown in Table 7.

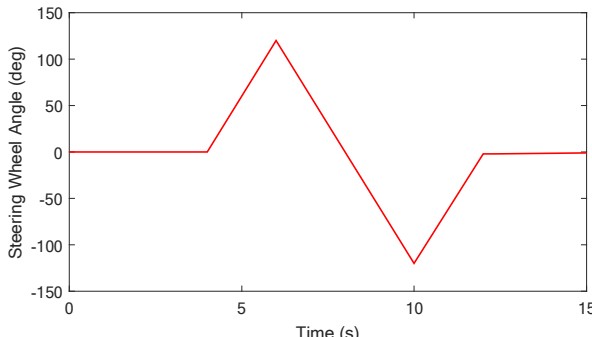

**Figure 16.** Steering wheel angle at the sinusoidal line shift condition.

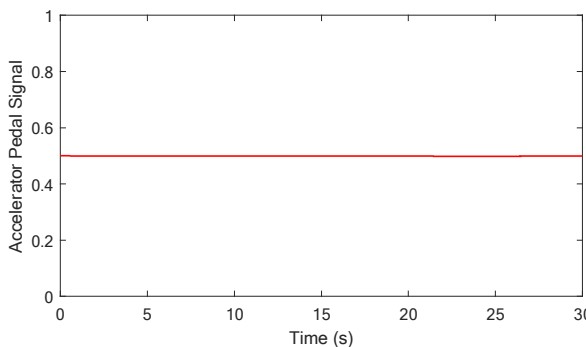

**Figure 17.** Acceleration pedal at the sinusoidal line shift condition.

As shown in Figure 18, the longitudinal speed of the vehicle in this condition is basically unchanged. Under the low adhesion road surface, the steering ability brought by the steering wheel angle is weakened, and the vehicle easily slides to the outside. The outside torque is greater than the inside torque, which can help the vehicle increase the steering ability. As shown in Figure 18 and Table 7, for vehicles without any control strategy, the yaw rate and sideslip angle increase sharply at 10 s, indicating that the vehicle has become unstable, and rollover occurs. Under sliding mode control and self-correcting fuzzy control, the actual values of the yaw rate and sideslip angle can follow the expected values well. In the vicinity of 7.5 s and 11.5 s, the deviation between the actual value and the expected value of the sideslip angle under the two control algorithms is large, but the maximum sideslip angle under the self-correcting fuzzy control strategy is smaller than the sliding mode control. Moreover, the maximum sideslip angle under self-correcting fuzzy control is 2.22 degrees, which is closer to the expected sideslip angle than the maximum sideslip angle under sliding mode control, and the vehicle trajectory tracking effect is better.

The simulation results show that, in the sinusoidal line-shifting condition, the vehicle runs at a medium speed and a small angle, and the vehicle rollover occurs when there is no control. The sliding mode control strategy and the self-correcting fuzzy control strategy have the same output torque trend of the same side wheel and maintain the same control trend of the yaw rate, sideslip angle of the center of mass and side acceleration. Both can reduce the lateral acceleration, realize differential control and restrain the roll and oversteer. However, compared with the sliding mode control, the self-correcting fuzzy control can reduce the maximum yaw rate deviation by 10% and the maximum sideslip angle deviation by 9.7%, indicating that the self-correcting fuzzy control strategy can effectively control the stability of vehicle running and steering. It is also proved that the fuzzy adjustment rule is effective in this working condition. The designed self-correcting fuzzy control can better track the desired trajectory and improve the vehicle's driving comfort and safety.

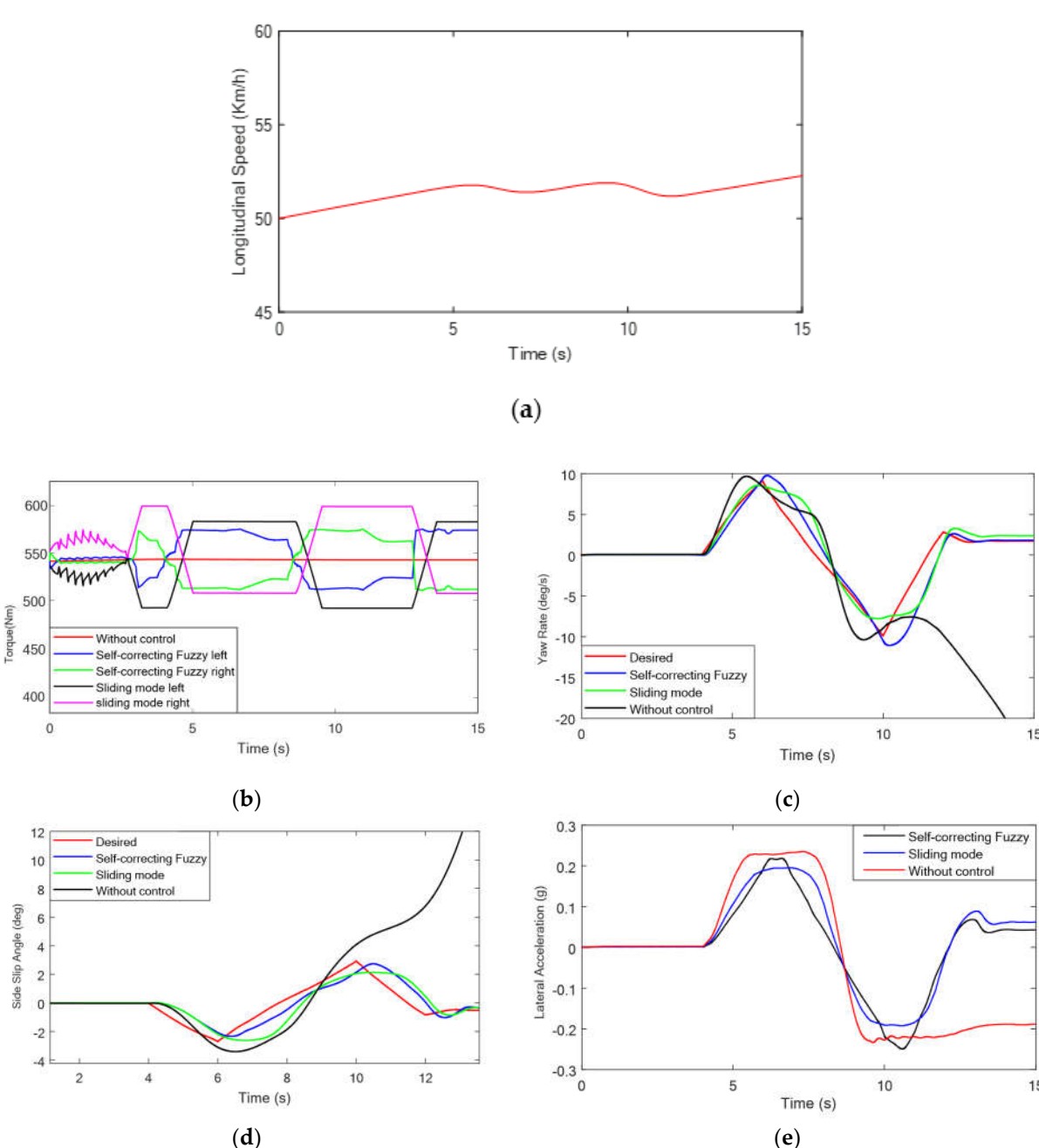

**Figure 18.** Simulation results of the distributed drive control at the sinusoidal line shift condition: (**a**) Speed; (**b**) Torque (**c**) Yaw Rate; (**d**) Side Slip Angle; (**e**) Lateral Acceleration.

**Table 7.** The maximum values of the main control parameters under the sinusoidal line shift condition.

| Control Strategy | Without Control | Sliding Mode | Deviation (%) | Self-Correcting Fuzzy | Deviation (%) |
|---|---|---|---|---|---|
| Maximum yaw rate (deg/s) | wandering | 8 | 20 | 11 | 10 |
| Maximum sideslip angle (deg) | wandering | 2 | 11 | 2.22 | 1.3 |

## 5. Conclusions

This paper takes a distributed rear drive electric bus as the research object. Based on the optimal distribution of the additional yaw moment, compare the control effects of two distributed control strategies, sliding mode control and self-tuning fuzzy control, on vehicle stability.

Using MATLAB and Trucksim software to build a seven-degrees-of-freedom bus model, the stability of the vehicle under three conditions—double shift line condition, high-speed small-steering-angle step condition and sinusoidal line shift condition—is simulated and analyzed. The final results show that both the sliding mode control and self-correcting fuzzy control can suppress the lateral acceleration of the vehicle, suppress the roll of the vehicle and also track the predetermined path. However, compared with the sliding mode control strategy, the self-correcting control strategy can better track the expected value of the yaw rate under the double shift line condition, and the deviation between the actual value and the expected value of the sideslip angle is reduced by 19%. Under the high-speed small-steering-angle step condition, the deviation between the actual value and the expected value of the yaw rate is reduced by 11%, and the deviation between the actual value and the expected value of the sideslip angle is reduced by about 6%. Under the sinusoidal shift line condition, the yaw rate can also better track the expected value, and the deviation between the actual value and the expected value of the sideslip angle is reduced by 9.7%. This shows that the self-correcting fuzzy control designed in this paper is better than the sliding mode control in the stability control of the electric bus, which can improve the tracking ability of the vehicle to the predetermined trajectory and improve the lateral stability of the vehicle.

In this paper, the effectiveness of the designed self-correcting fuzzy control strategy for vehicle stability control was verified through the co-simulation platform. However, the self-correction of the quantization factor and scale factor requires the control effect correction based on the system. However, when the control system changes, the control effect changes, and the correction coefficient needs to be reset. In future research, a hardware-in-the-loop experiment and a road vehicle test will be carried out, and self-correcting fuzzy control can be further studied to realize the function of automatically adjusting the correction coefficient when the control system is switched so as to achieve self-correction in the true sense.

**Author Contributions:** Conceptualization, methodology, H.C.; investigation, formal analysis, J.G. and W.W.; validation, writing—original draft preparation, W.L. and S.C.; supervision, writing—review and editing, H.C. and S.Z.; funding acquisition, R.Z. All authors have read and agreed to the published version of the manuscript.

**Funding:** We gratefully acknowledge the financial support of this research by the following projects: Control design of a new energy vehicle air conditioning compressor based on intelligent multi-objective optimization (Grant No. ZDLQ2020002), Research on the theory and method of the autonomous cooperative operation control of high-speed trains (Grant No. U1934221) and Research and manufacture of a high-precision grinding process of wear-resistant seals of construction machinery (Grant No. ZDLQ2021005).

**Data Availability Statement:** Not applicable.

**Conflicts of Interest:** The authors declare no conflict of interest.

## Nomenclature

| | | | | |
|---|---|---|---|---|
| $l_f$ | the distance from the center of mass of the vehicle to the front axles | | $l_r$ | the distance from the center of mass of the vehicle to the rear axles |
| $l$ | wheelbase | | $v_x$ | the longitudinal speed |
| $k_f$ | the cornering stiffness of the front axles of the vehicle | | $k_r$ | the cornering stiffness of the rear axles of the vehicle |
| $I_Z$ | the moment of inertia of the vehicle around the Z axis | | $\delta_F$ | the front wheel steering angle |
| $K$ | stability factor | | $\mu$ | road adhesion coefficient |
| $r$ | yaw rate | | $\beta$ | sideslip angle |
| $r_{bound}$ | the boundary value of the yaw rate | | $\beta_{bound}$ | the boundary value of the sideslip angle |
| $r_d$ | the expected value of the yaw rate | | $\beta_d$ | the expected value of the sideslip angle |
| $e_r$ | the deviation between the actual value and the expected value of the yaw rate | | $e_\beta$ | the deviation between the actual value and the expected value of the sideslip angle |
| $\varphi$ | vehicle yaw | | $R$ | the wheel rolling radius |
| $F_{x_F}$ | the longitudinal reaction forces of the front driving wheels | | $F_{x_R}$ | the longitudinal reaction forces of the rear driving wheels |

| $F_{y_F}$ | the lateral reaction forces of the front driving wheels | $F_{y_R}$ | the lateral reaction forces of the rear driving wheels |
|---|---|---|---|
| $I_w$ | the rotational inertia of the wheel | $F_d$ | the wheel friction |
| $T_d$ | the wheel driving torque | $T_b$ | the wheel braking torque |
| $T_{RL}$ | the driving torque of left rear wheels | $T_{RR}$ | the driving torque of the right rear wheels |
| $m$ | vehicle mass | $h$ | height of the center mass |
| $w$ | wheel pitch | $\Delta M$ | additional yaw moment |
| $s$ | the sliding mode variable | $\lambda$ | the weight coefficient |
| $c_r$ | the relative weight coefficient between the yaw rate deviation and the derivative | $K_v$ | constant approaching rate |
| $\Delta_r$ | the boundary layer thickness parameter | $K_{pd}$ | pedal opening degree |
| $K1$ $K2$ | the proportional factors | $K3$ | the scale factors of the output variable |
| $\delta1$ $\delta2$ $\delta3$ | the correction coefficients | $T(*)$ | the fuzzy set domain |
| $\mu_{A_r^i}(*)$ $\mu_{A_r^i}(*)$ $\mu_{B^i}(*)$ | membership functions | $d(K1_i)$ $d(K2_i)$ $d(K2_i)$ | self-correcting variable |
| DYC | direct yaw moment control | NB | negative big |
| NM | negative medium | NS | negative small |
| ZE | zero | PS | positive small |
| PM | positive medium | PB | positive big |
| HIL | hardware-in-the-loop | | |

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
