# Peer review of "Design of a Wheel-Side Rear-Drive Distributed Electric Bus Control Strategy Based on Self-Correcting Fuzzy Control"

_electronics, doi:10.3390/electronics11244219_

Round 1
Reviewer 1 Report
In this paper, the authors would like to effectively utilize the advantage of the independent control ability of each rear wheel, they designed and compared two direct transverse moment control strategies based on sliding mode control and self-correcting fuzzy control, and distributed the drive torque in combination with the vehicle steering torque constraint. They also provided a serious experiment. The results showed that compared with the sliding mode control, the self-correcting fuzzy control strategy can reduce the maximum sideslip angle deviation by 19%, 6%, and 9.7% respectively under the double shift line condition, the high-speed small steering angle step condition and the sinusoidal line shift condition, and can more effectively reduce the vehicle lateral acceleration and improve the vehicle yaw rate tracking ability, significantly improve the lateral stability of the vehicle.
The authors have presented an improved version of their paper. This paper is well-written and has the potential to be published in the Journal of Electronics. However, I have some issues that need to be improved in this paper before which be published in such a highly-ranked journal.
[1] The reference can be improved by adding more recent relative topics to your research work.
[2] In Equation (13), you mentioned that sat (s) is transferred from sgn(s), but we can’t know what is sgn(s) and how to transfer it into sat(s).
Reviewer 2 Report
1. The phenomenon of self-correcting fuzzy control is not clear in the paper.
2. How are the authors adjusting the rules during the fuzzy control operation?
3. Why only self-correcting fuzzy is chosen in the research why not some advanced and proven highly efficient rule growing scenarios or online learning approaches for establishing independent control?
4. The novelty and major contributions of the manuscript should be clearly presented in the introductions (Preferably in bullet points).
5. Literature needs to be improved and the drawbacks of the literature in-terms of their capability in solving the problem identified should be addressed. General controller drawbacks are not an ideal drawback for using the control algorithms developed in the research.
6. Are the model developments in Section 2 originally derived by the authors, or adapted from the literature? Please cite the relevant literature?
7. Why did the authors only consider a two degree of freedom model (Figure 4) while designing the distributed drive control?
8. The equations (18)-(24) are widely established. Please cite the relevant references.
9. Please show the drive model and tabulate the vehicle parameters used in the simulation.
10. The explanation provided for the results is very brief and a comparison with the results of the problem solved in the literature is necessary.
